

# Analysis of five complete genome sequences for members of the class Peribacteria in the recently recognized Peregrinibacteria bacterial phylum

Karthik Anantharaman[1], Christopher T. Brown[2], David Burstein[1], Cindy J. Castelle[1], Alexander J. Probst[1], Brian C. Thomas[1], Kenneth H. Williams[3] and Jillian F. Banfield[1,3]

[1] Department of Earth and Planetary Sciences, University of California, Berkeley, California, United States
[2] Department of Plant and Microbial Biology, University of California, Berkeley, California, United States
[3] Earth Sciences Division, Lawrence Berkeley National Laboratory, Berkeley, California, United States

Corresponding author
Jillian F. Banfield,
jbanfield@berkeley.edu

## ABSTRACT

Five closely related populations of bacteria from the Candidate Phylum (CP) Peregrinibacteria, part of the bacterial Candidate Phyla Radiation (CPR), were sampled from filtered groundwater obtained from an aquifer adjacent to the Colorado River near the town of Rifle, CO, USA. Here, we present the first complete genome sequences for organisms from this phylum. These bacteria have small genomes and, unlike most organisms from other lineages in the CPR, have the capacity for nucleotide synthesis. They invest significantly in biosynthesis of cell wall and cell envelope components, including peptidoglycan, isoprenoids via the mevalonate pathway, and a variety of amino sugars including perosamine and rhamnose. The genomes encode an intriguing set of large extracellular proteins, some of which are very cysteine-rich and may function in attachment, possibly to other cells. Strain variation in these proteins is an important source of genotypic variety. Overall, the cell envelope features, combined with the lack of biosynthesis capacities for many required cofactors, fatty acids, and most amino acids point to a symbiotic lifestyle. Phylogenetic analyses indicate that these bacteria likely represent a new class within the Peregrinibacteria phylum, although they ultimately may be recognized as members of a separate phylum. We propose the provisional taxonomic assignment as 'Candidatus Peribacter riflensis', Genus Peribacter, Family Peribacteraceae, Order Peribacterales, Class Peribacteria in the phylum Peregrinibacteria.

## INTRODUCTION

Metagenomic analyses of microbial communities in natural environments have revealed numerous previously unrecognized microorganisms, including those now described to be

part of the phylum Peregrinibacteria. The Peregrinibacteria are a lineage within the bacterial CPR (*Wrighton et al., 2012*; *Brown et al., 2015*), a monophyletic radiation primarily of candidate phyla. Members of the Peregrinibacteria phylum have been genomically sampled from a contaminated sediment-hosted aquifer (*Wrighton et al., 2012*; *Brown et al., 2015*) and a deep subsurface high $CO_2$ environment (*Emerson et al., 2015*). However, ribosomal proteins and 16S rRNA gene sequences from this group have been detected in a variety of environments, including sediments and soil (*Hug et al., 2015*). Presently, there are only 18 partial genomes from this phylum. Prior analyses (*Wrighton et al., 2012*) predicted the lack of a complete respiratory chain and most TCA cycle components, consistent with a fermentation-based lifestyle. However, lack of complete, closed genomes from this phylum have limited confidence about predicted metabolic potential and genome size.

Here, we report the first complete genomes from the Peregrinibacteria bacterial phylum. These genomes represent five very closely related strains of a single species.

## MATERIALS AND METHODS

### Sampling

Groundwater from an alluvial aquifer near the town of Rifle, Colorado (Latitude 39.52927920, Longitude −107.77162320, altitude 1618 m above sea level (*Williams et al., 2011*)) was pumped from a depth of 5 m below ground surface. Samples were collected prior to injection of oxygen-saturated water up gradient from the sampling wells (August 2, 2012) and on three dates after the injection (September 12, 2012, October 24, 2012 and December 12, 2012) and filtered through sequential 1.2 μm, 0.2 μm and 0.1 μm filters.

### DNA sequencing, assembly, binning and annotation

DNA was extracted from the filters, as described previously (*Brown et al., 2015*), and paired-end libraries with a read length of 150 bp were sequenced using Illumina HiSeq 2000 at the Joint Genome Institute. Sequence data were processed using the Illumina CASAVA pipeline (version 1.8). Reads were trimmed using Sickle (https://github.com/najoshi/sickle) and assembled using IDBA-UD (-step 20, -maxk 100, -mink 40) (*Peng et al., 2012*). All samples were assembled separately (Supplementary Table S1). Binning of assembled scaffolds was performed on the basis of GC content and differential read coverage with ABAWACA (*Brown et al., 2015*), and tetranucleotide composition with Emergent Self-Organizing Maps (ESOM) (*Dick et al., 2009*). The five genomes described in this study were recovered from five separate samples (A2, B2, C2, D1, D2) across all four sampling time points (A, B, C, D) (Table 1).

Subsequently, individual bins were inspected for scaffolding errors and identified scaffolds were reassembled using Velvet (*Zerbino, 2010*) as described previously (*Brown et al., 2015*). Read-mapping was performed using Bowtie2 (*Langmead & Salzberg, 2012*) with default parameters. Visualization of mapped reads and manual curation of the genomes was conducted in Geneious (*Kearse et al., 2012*) to check the integrity of the five assembled genomes. Complete circularization and filling of internal gaps was achieved using mapped reads from the same sample in three cases, and in two

**Table 1 Summary of the five *Ca.* P. riflensis genomes.**

| Genome | Source sample | Length (bp) | GC (%) | Coverage (x) | ORFs |
|---|---|---|---|---|---|
| RIFOXYA2_FULL_PER-ii_58_14 | A2 | 1,248,026 | 58.48 | 14 | 1,102 |
| RIFOXYB2_FULL_PER-ii_58_17 | B2 | 1,248,181 | 58.48 | 23 | 1,104 |
| RIFOXYC2_FULL_PER-ii_58_32 | C2 | 1,248,112 | 58.48 | 32 | 1,101 |
| RIFOXYD1_FULL_PER-ii_59_16 | D1 | 1,247,854 | 58.48 | 18 | 1,101 |
| RIFOXYD2_FULL_PER-ii_51_23 | D2 | 1,248,180 | 58.48 | 27 | 1,105 |

cases, from a different sample. Protein-coding genes were predicted using Prodigal (*Hyatt et al., 2010*), and rRNA genes using Rfam (*Nawrocki et al., 2015*). The genomes were annotated by searching the predicted proteins against Uniref90 (*Suzek et al., 2007*), KEGG (*Kanehisa & Goto, 2000*) and a custom in-house database (*Brown et al., 2015*) using Usearch (-ublast) (*Edgar, 2010*). Functional predictions for proteins of interest were further analyzed using Interpro (*Jones et al., 2014*), psortb (*Yu et al., 2010*), phylogenetic analysis and HMM models constructed from KEGG proteins assigned the same KO annotation (9), which were further sub-clustered using MCL (I = 1.1) (*Van Dongen, 2008*) based on full-length identity percent.

Analysis of Single-Copy Genes (SCG) was performed using 43 universal marker genes (*Raes et al., 2007*; *Brown et al., 2015*).

## Phylogenetic analysis

Phylogenetic analysis for validating the taxonomy of the 5 Peregrinibacteria strains was performed with the 16S ribosomal RNA (SSU) gene as well as a set of 16 syntenic universal Ribosomal Proteins (RP) (L2, L3, L4, L5, L6, L14, L15, L16, L18, L22, L24 and S3, S8, S10, S17, S19).

For SSU analysis, 69 SSU genes from the phylum Peregrinibacteria were aligned with MUSCLE (*Edgar, 2004*) with default parameters. All introns in 16S rRNA genes were removed as described previously (*Brown et al., 2015*). Sequences from the phylum Berkelbacteria were used as an outgroup. All columns with >97% gaps were removed and the final alignment spanned 1,649 nucleotides. Phylogenetic analysis of SSU genes was inferred by RAxML using the GTRGAMMA algorithm with 1000 bootstrapped replicates (*Stamatakis, 2014*).

RAxML was called as follows: raxmlHPC-PTHREADS -# 1000 -s input -n output -m GAMMAGTR -p 777 -x 3333 -f a.

For the RP analysis, each ribosomal protein was aligned individually along with reference sequences using MUSCLE (*Edgar, 2004*) with default parameters. The Reference sequence database comprised proteins from 11 previously identified Peregrinibacteria, and Berkelbacteria (*Wrighton et al., 2012*; *Brown et al., 2015*) that served as an outgroup. Individual ribosomal protein alignments were concatenated in Geneious version 7 (*Kearse et al., 2012*). All columns with >97% gaps were removed prior to further analyses. In total, the alignment spanned 3085 columns. Phylogenetic analysis of RP was inferred by

RAxML using the PROTGAMMAGTR algorithm with 100 bootstrapped replicates (*Stamatakis, 2014*). RAxML was called as follows: raxmlHPC-PTHREADS -# 100 -s input -n output -m PROTGAMMAGTR -p 777 -x 777 -f a.

Phylogenetic trees were visualized with Figtree v1.2.2 (http://tree.bio.ed.ac.uk/software/figtree/).

## RESULTS AND DISCUSSION

Three complete, circularized genomes were reconstructed from samples D2, B2 and C2. For two other genomes (from samples A2 and D1), very small gaps were filled using mapped reads from other samples. All genomes contain one direct repeat region of a length similar to that of the read insert size. The correct assembly path was chosen because the alternative would have excluded an ~3 kbp sequence that must be part of the genome because paired reads placed there.

The average length of the complete genomes was 1,248,071 ± 217 bp, a result that confirms the previous prediction of small genome size that was made based on genome completeness estimates and draft genome lengths (*Wrighton et al., 2012*). The average GC content was 58.5% and average read coverage varied from $14\times$ to $32\times$ (Table 1). A plot of GC skew across the genome has the expected form, providing confidence in the accuracy of the assembly and an indication of the locations of the origin and terminus of replication (which are not otherwise apparent from the gene predictions). We predicted between 1,101 and 1,105 coding sequences. 52% of the genes had confident functional assignments, based on HMM analysis (Supplementary Table S2), 29% additional proteins had weak similarity to the HMM database (below predetermined noise cutoff), and the others had no detectable similarity to any HMM (19% of all genes in the genome). Notably, 29% of the genes had no significant BLAST hit against NCBI's nr database (E-value $<10^{-5}$, percent identity >20%). We identified 46 tRNA genes, single copies of each of the 5S rRNA, 16S rRNA and 23S rRNA genes, and all 43 universal SCG in the five genomes. No insertion sequences, such as reported in many CPR bacteria, were identified in the 16S rRNA genes (*Brown et al., 2015*). The 16S rRNA gene sequences of all five genomes are identical, suggesting they derive from strain variants of a single species. The 16S rRNA gene sequence is only 89% similar to that of the closest genomically-sampled relative, Peregrinibacteria bacterium GW2011_GWC2_54_8 (LCRS00000000) (Supplementary Table S3) (*Brown et al., 2015*). The sequences from the genomes reported here, those of GWC2_PER_54_8 and a highly similar genotype GWB1_PER_54_8, and clone sequences identified in the SILVA database (*Pruesse et al., 2007*), branch separately from the other Peregrinibacteria for which partial genomes are available. Thus, the newly reported genomes likely represent a new class within the Peregrinibacteria (Fig. 1). It remains to be seen whether this lineage should instead be recognized as a separate phylum sibling to the Peregrinibacteria. We propose the name Peribacteria for this class. We designate the species as 'Candidatus Peribacter riflensis' and suggest this lineage be placed as Genus Peribacter, Family Peribacteraceae, Order Peribacteria, Class Peribacteria, within the phylum Peregrinibacteria.
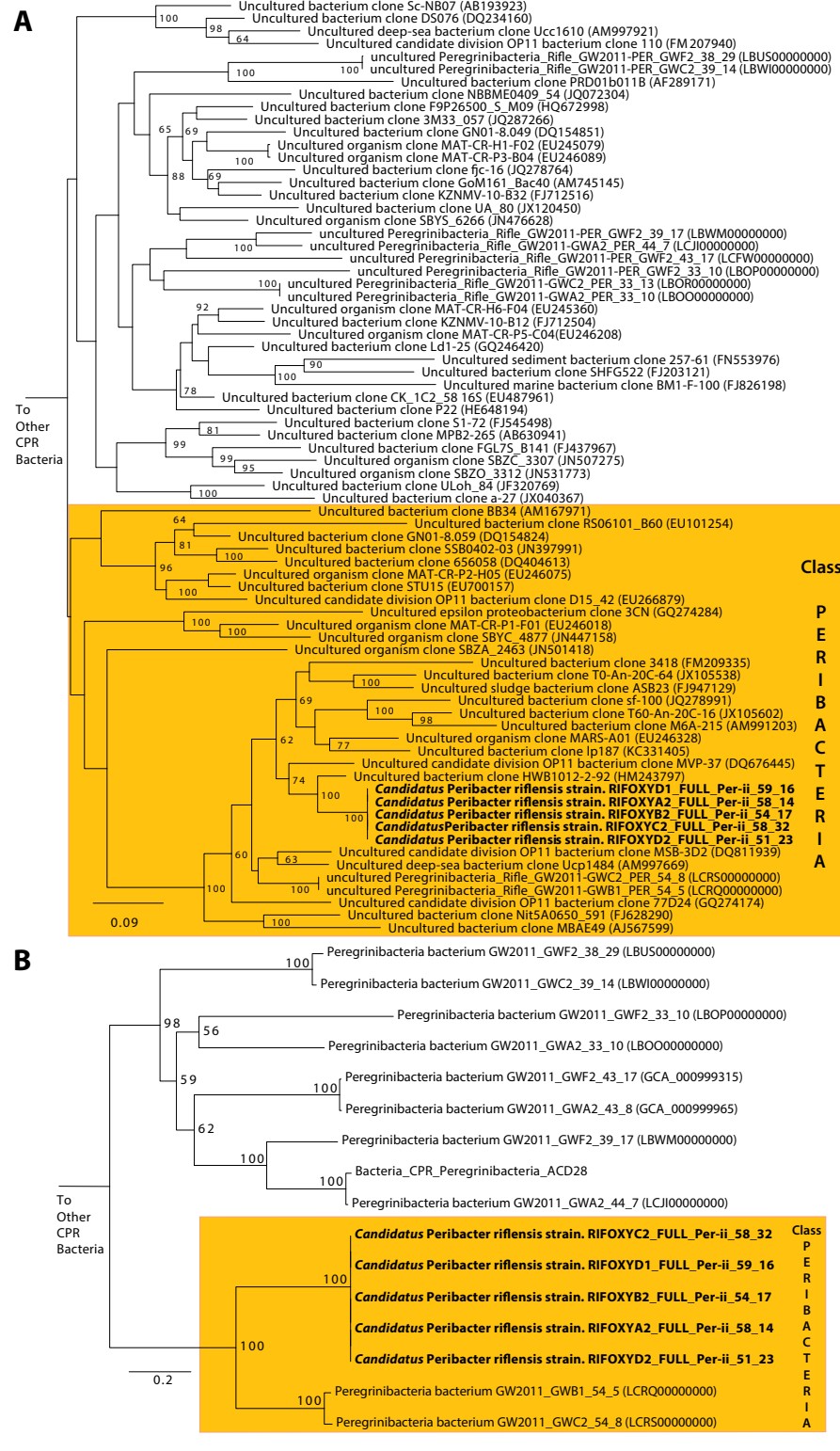

**Figure 1 Phylogenetic placement of the *Ca.* P. riflensis genomes.** (A) 16S rRNA and (B) concatenated ribosomal protein trees showing the phylogenetic placement of the *Ca.* P. riflensis genomes inferred by maximum likelihood. Bootstrap values (percent of 1000 bootstrap repetitions) are marked on internal nodes. Only bootstrap values ≥50 are shown.

Variation among the five *Ca.* P. riflensis genomes primarily involves insertions and deletions (tens of bps) in twelve proteins (Fig. 2). Of these, seven proteins were very large (between 1,219 and 3,861 amino acids) and several were predicted to have an extracellular localization.

A metabolic reconstruction for the *Ca.* P. riflensis species was established based on analysis of annotated genes (Fig. 3). Examination of the genomes revealed few glycosyl hydrolases that would support growth by utilization of externally derived sugar polymers, a lifestyle suggested for many other CPR bacteria (*Wrighton et al., 2012*; *Kantor et al., 2013*). We identified pathways for both formation and utilization of glycogen/starch and dextrin. Externally-derived glycogen/starch and dextrin-like compounds may be broken down using these enzymes to provide inputs to glycolysis. In addition, the genomes encodes seven enzymes lacking confident functional predictions, but with weak similarity to dextrinases involved in the hydrolysis of starch-derived dextrin. The genomes also encode for synthesis of the compatible solute trehalose.

The genomes lack genes encoding for a complete TCA cycle and most components of respiratory chains, although those for the subunits of the ATP synthase complex were identified, along with an inorganic pyrophosphatase, providing a route to make ATP. The genomes encodes a protein annotated as cytochrome oxidase subunit II, as do those of many other CPR (*Brown et al., 2015*), but the catalytic subunit I that is critical for $O_2$ utilization is not present. Thus, the ability to metabolize $O_2$ is not supported.

We predict that the conversion of alpha-D-glucose to pyruvate occurs via the upper and lower glycolytic pathway, with the lack of 6-phosphofructokinase compensated for by enzymes of the Pentose Phosphate Pathway (PPP). This workaround is a common strategy in many members of the CPR that lack 6-phosphofructokinase (*Wrighton et al., 2012*). Lacking are genes required for gluconeogenesis (fructose-1,6-bisphosphatase, pyruvate carboxylase and PEP carboxykinase were not identified). Pyruvate can be converted to oxaloacetate (via phosphoenolpyruvate carboxylase) and to acetyl-CoA (via the two subunit Pyruvate:Ferredoxin Oxidoreductase Complex (PFOR)). The genes for ferredoxin and Fe-S cluster biosynthesis are encoded, as is a gene involved in synthesis of flavodoxin. The redox recycling of ferredoxin is unclear, but one subunit of a Fe-hydrogenase (ferredoxin hydrogenase large subunit) may be involved (however other subunits were not identified). A pathway involving electron bifurcation may be required for production of reduced ferredoxin, possibly involving an electron transferring flavoprotein, but candidate mechanisms are not apparent. One potential mechanism may involve the oxidation of alanine through a Stickland reaction, which could result in oxidative phosphorylation to form ATP and NADH. In turn, the generated NADH could serve as the substrate for electron bifurcation to produce reduced ferredoxin.

We identified genes for production of cofactors needed in nucleotide and other metabolism. Although the genes for biosynthesis of folate itself were not identified, enzymes for tetrahydrofolate compounds (required for one carbon transfer reactions during biosynthesis of serine, methionine, glycine, and purine nucleotides, for example) are present. Folate may be acquired from other organisms, as is often the case for this cofactor (*Graber & Breznak, 2005*). We identified a complete biosynthesis pathway for

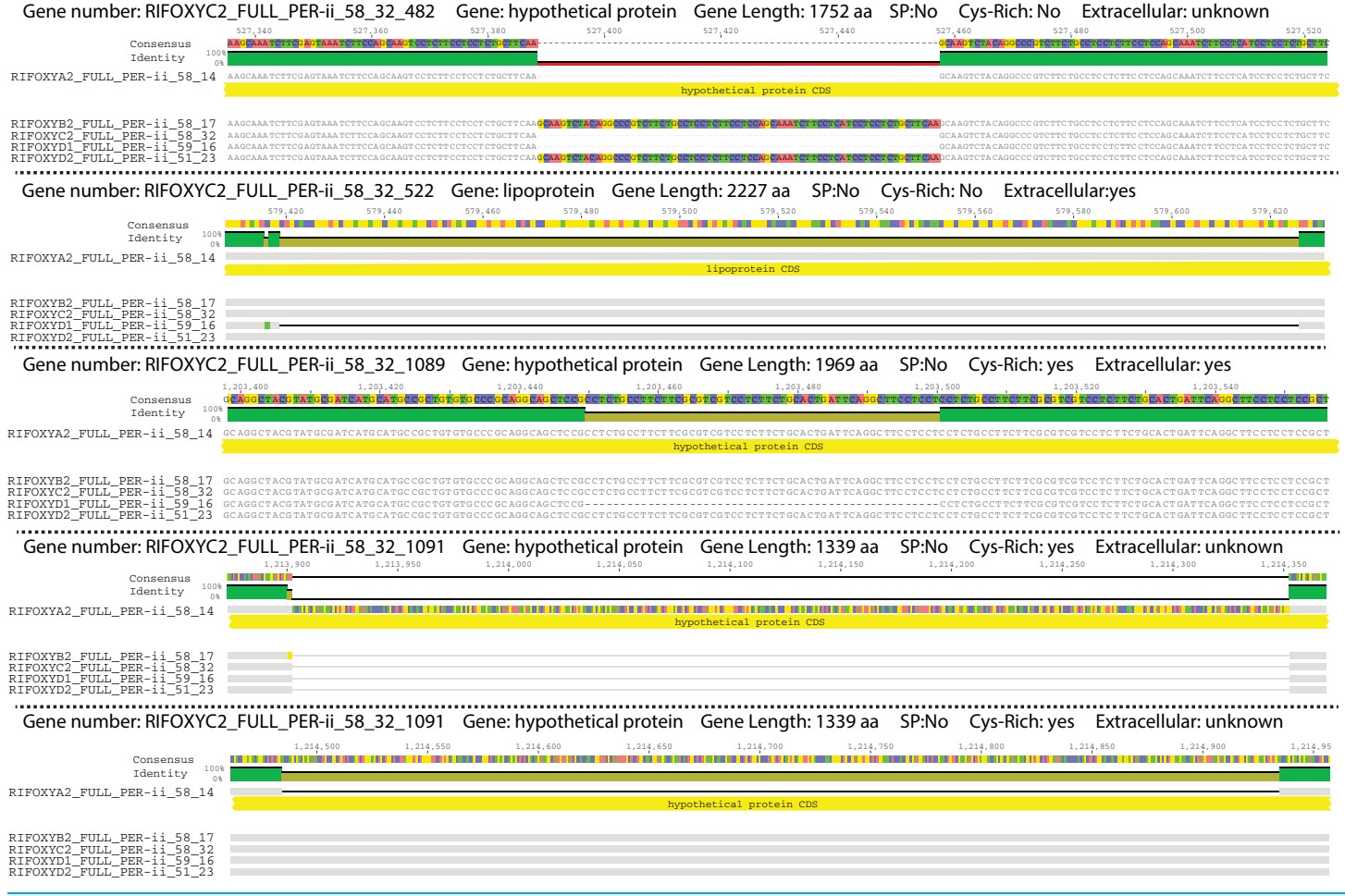

**Figure 2 Diagram showing the form of variation that distinguishes the five closely related genomes.** Coding Region on the genome is shown in yellow. Colors on the identity bar reflect similarity between the five genomes: Green, 100%; Olive, 80%; Red, 40%; Black, 20%.

riboflavin and its conversion to FMN and possibly to FAD. Genes for biotin and thiamine biosynthesis were not identified, and are probably not needed due to the absence of pathways in which these enzymes are required (e.g., fatty acid biosynthesis and pyruvate carboxylase of the gluconeogenesis pathway). The source of NAD, a cofactor in several proteins in these genomes (e.g., $NAD^+$-dependent epimerase), is unclear. However, $NAD^+$ kinase for interconverting $NAD^+$ and $NADP^+$ and ADP-dependent NAD(P)H-hydrate dehydratase for dehydration of NADHX and NADPHX were identified.

Genes of the PPP also encode enzymes to form D-ribose 5-phosphate and 5-phospho-alpha-D-ribose 1-diphosphate that are required for synthesis of nucleotides for use as energy carriers and for production of purine, pyrimidine and RNA. In fact, essentially complete pathways for nucleotide synthesis are encoded in the *Ca.* P. riflensis genomes. This capacity distinguishes these organisms from most other CPR bacteria, but it may be a shared feature of some Peregrinibacteria. Genes for homologous recombination (Holliday junction Ruv pathway) and *recA* were also identified, as were genes for cell division and DNA repair. The origin and terminus of replication were identified using a GC skew plot

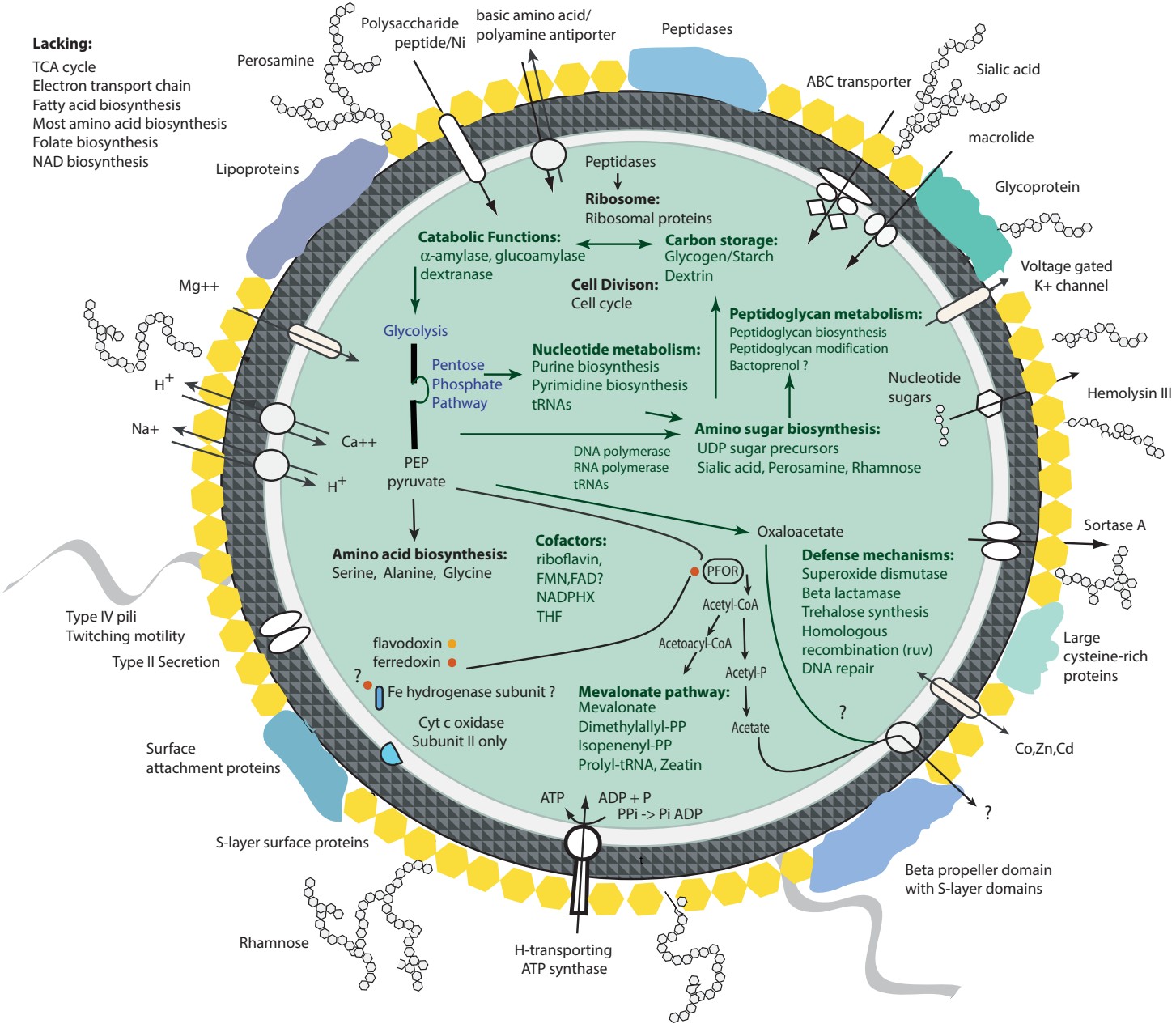

**Figure 3 Overview of the protein investments in biosynthesis in *Ca.* P. riflensis.** TCA, Tricarboxylic acid cycle; NADPHX, Reduced nicotinamide adenine dinucleotide phosphate; FMN, Flavin mononucleotide; FAD, flavin adenine dinucleotide; PEP, phosphoenolpyruvate; THF, Tetra-hydrofolic acid; PPi, Pyrophosphate; PFOR, pyruvate:ferredoxin oxidoreductase.

(Fig. 4A), which shows the form expected for a correctly assembled genome. Short repeats were localized close to the replication origin, and DNaA box-like sequences were present in the vicinity (Fig. 4B).

Unlike some other Peregrinibacteria, *Ca.* P. riflensis lack form II/III RuBisCO, which has been implicated in $CO_2$ incorporation using a nucleotide salvage pathway (*Wrighton et al., 2012*). Also lacking are hydrogenases (other than the above mentioned single subunit) and CRISPR-Cas phage defense.

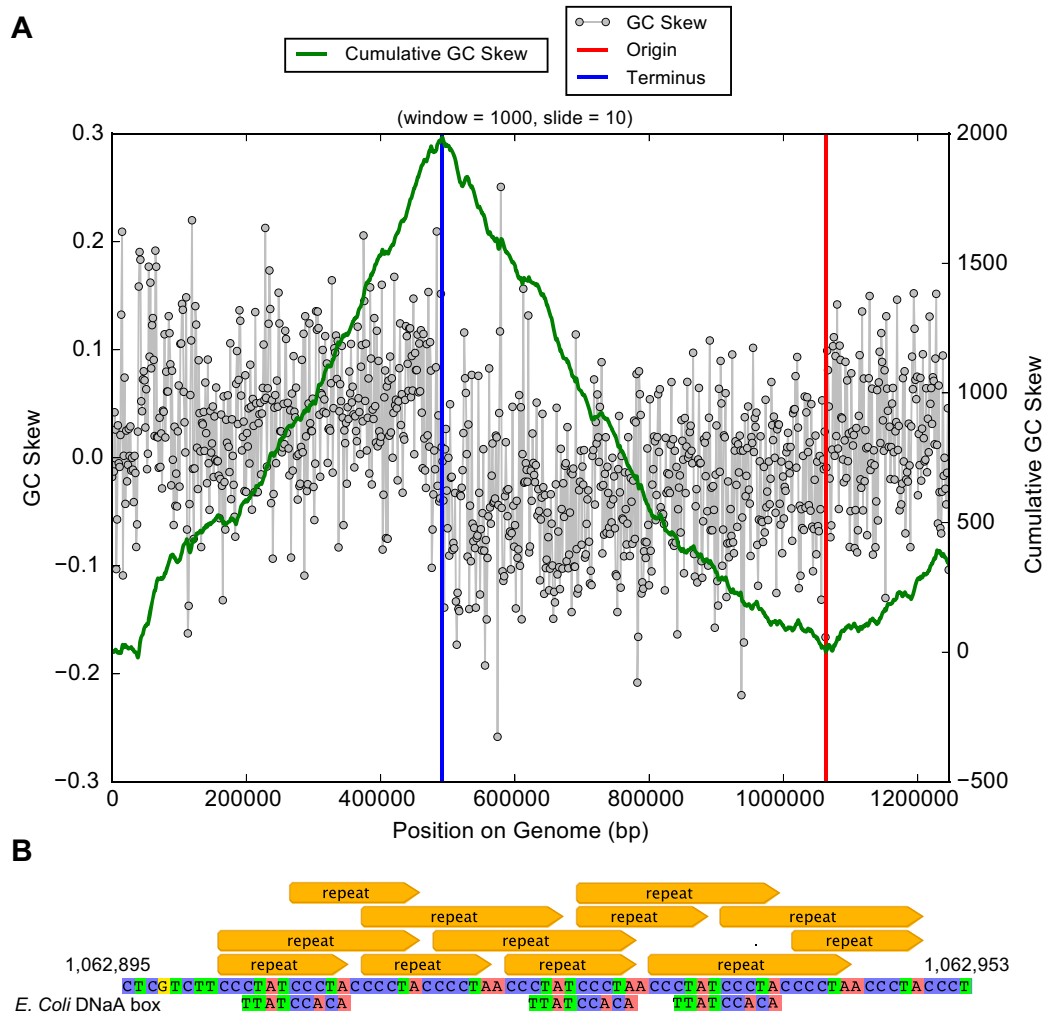

**Figure 4 Plot of GC skew over the *Ca.* P. riflensis str.RIFOXYC2_FULL_PER-ii_58_32 genome.**
(A) Plot of GC skew indicating the locations of the origin/terminus of replication. The plot has the expected form, providing added confidence about the accuracy of the assembly. (B) Localization of short repeats in the identified origin of replication. *E. Coli* DNaA box sequences are shown for reference.

There are only few genes predicted to be involved in amino acid biosynthesis. Synthesis of alanine, serine and glycine is possible. The genomes encode a wide variety of peptidases (including methionyl aminopeptidase, carboxyl-terminal processing protease, Xaa-Pro dipeptidase, ATP-dependent Clp protease). Thus, we infer that these bacteria obtain amino acids via degradation of externally-derived proteins. We identified relatively few genes for degradation of amino acids (e.g., serine hydroxymethyl transferase, L-asparaginase and aspartate amino transferase) to ammonia, oxaloactetate or pyruvate. Thus, we conclude that externally derived amino acids (rather than their breakdown products) are key to protein biosynthesis.

We did not identify a gene for production of acetate from acetyl-CoA. This function is often encoded in CPR genomes as a way to dispose of acetyl-CoA and to produce ATP via the mechanism of substrate-level phosphorylation as part of a fermentation-based metabolism. Acetate production may occur via a pathway involving a protein identified as

having a domain that has been implicated in conversion of acetyl-CoA to acetyl-$P_i$ (an acetyltransferase), and a confidently identified enzyme identified to convert acetyl-$P_i$ to acetate (with production of ATP).

Conversion of acetyl-CoA to acetoacyl-CoA is possible. This is likely used in isoprenoid biosynthesis as the genes of the mevalonate pathway, including those that form mevalonate and mevalonate-5P and those to convert mevalonate-5PP to isopentenyl-PP and dimethylallyl-PP were identified. This pathway is relatively rare in bacteria (*Lombard & Moreira, 2010*) outside of the CPR. The purpose for isoprenoids formed by this pathway is uncertain. Intriguingly, however, one gene encoding for an enzyme required to convert dimethylallyl-PP to a zeatin (a plant hormone) precursor was identified (tRNA dimethylallyltransferase).

The source of bacterial fatty acids required for construction of the cytoplasmic membrane is also unclear, as only $\beta$-ketoacyl-Acyl Carrier Protein (ACP) reductase (*fabG*) was identified. The many genes for biosynthesis of isoprenoid-based compounds suggest the importance of these molecules in the cell envelope, although there is no indication that they are the predominant structural constituents (*Luef et al., 2015*).

All genes for biosynthesis and modification of peptidoglycan were identified, including genes for production of the amino sugar precursors. Genes for synthesis of cell surface-associated sugar-containing molecules including polysaccharides and a variety of amino sugars were identified, including those for production of sialic acid. In addition, we identified genes encoding for aquaporin Z. The presence of hydrophilic polymers and aquaporin may be important for life in groundwater, where these cells are disproportionately highly abundant (relative to sediment) (*Hug et al., 2015*). Also present are genes for production of rhamnose (dTDP-l-rhamnose synthase is encoded by *rfbABCD*) and perosamine, a mannose-derived 4-aminodeoxysugar. Notably, the genomes encode three copies of perosamine synthetase (*rfbE*) and two copies of glucose-1-phosphate thymidylyltransferase (*rfbA*) suggesting the importance of cell surface polymers to these bacteria. Also present in at least four copies is phosphatidylglycerol: prolipoprotein diacylglycerol transferase. The function of these enzymes in *Ca*. P. riflensis is unclear, but they may be involved in modification of surface proteins, including lipoproteins. Also identified in multiple copies (possibly 8) are proteins with the annotation of dichol-phosphate mannosyltransferase. These may be involved in sugar modification of polyprenols (possibly formed by the mevalonate pathway).

An interesting feature of the Peribacteria genomes is the relatively high cysteine content (1.67%) of the predicted proteins. This value is significantly larger than those reported for bacteria from aquatic and terrestrial environments (average values of 0.05% and 0.15%, although these previously published data indicate substantial within-environment variation) (*Moura, Savageau & Alves, 2013*). Notably, 19% of these cysteines are incorporated into twenty proteins including a 14-gene cluster (7.2 ± 2.4% cysteine) (Supplementary Table S4). These C-rich proteins are large (average length 1,655 ± 751 aa), most are predicted to have a signal peptide and many have predicted extracellular localization, although some may be associated with the cytoplasmic membrane. Two are annotated as integrin alpha FG-GAP repeat containing protein 1, one as a lactonase;

the remainder lack functional predictions, but several show very weak similarity to *Myxococcus* cysteine-rich repeat proteins (and to pappalysin-1). One protein for disulfide bond formation was present. The very high cysteine content may contribute to protein rigidity and/or to surface attachment (*Nakayama, Kurokawa & Lee, 2012*). Genes for production of S-layer proteins were also identified. Other predicted exported proteins include some with lipoprotein annotations (e.g., *NlpD*), and a sortase A. Sortases occur in Gram-positive bacteria, where they function in export and attachment of surface proteins to the cell wall (*Mazmanian, Ton-That & Schneewind, 2001*). Also encoded are genes for twitching motility, genes involved in sensing and response (chemotaxis), a Type II secretion system and genes for biosynthesis of type IV pili, possibly involved in inter-organism interactions (*Luef et al., 2015*).

In summary, these genomes are notable because of the high incidence of genes that encode large exported and often cysteine-rich proteins, substantial investment in production of amino sugar-based polymers, a near-complete mevalonate pathway for production of isoprenoids, and multi-copy genes possibly involved in modification of exported proteins with lipids and sugars. These features point to a prioritization towards synthesis of cell surface-associated compounds by these bacteria. The metabolic predictions are consistent with, at best, a semi-independent lifestyle. Degradation pathways funnel carbon into acetyl-CoA, oxaloacetate, and possibly acetate. Given scant opportunities for use of oxaloacetate by these bacteria, it is possible that this important carbon currency is exported for use by host bacteria. If *Ca.* P. riflensis are symbionts, the localization of sequence variation in large exported proteins may reflect adaption of the different strains to distinct host species.

## ACKNOWLEDGEMENTS

DNA sequencing was conducted at the DOE Joint Genome Institute, a DOE Office of Science User Facility, via the Community Science Program.

### Funding

This work was supported by Lawrence Berkeley National Laboratory's Sustainable Systems Scientific Focus Area funded by the U.S. Department of Energy, Office of Science, Office of Biological and Environmental Research under contract DE-AC02-05CH11231. DB was supported by a long-term EMBO fellowship. DNA sequencing was conducted at the DOE Joint Genome Institute, a DOE Office of Science User Facility, via the Community Science Program. The funders had no role in study design, data collection and analysis, decision to publish, or preparation of the manuscript.

### Grant Disclosures

The following grant information was disclosed by the authors:
U.S. Department of Energy, Office of Science, Office of Biological and Environmental Research: DE-AC02-05CH11231.

## Competing Interests
The authors declare no conflict of interest.

## Author Contributions
- Karthik Anantharaman conceived and designed the experiments, performed the experiments, analyzed the data, contributed reagents/materials/analysis tools, wrote the paper, prepared figures and/or tables, reviewed drafts of the paper.
- Christopher T. Brown performed the experiments, analyzed the data, contributed reagents/materials/analysis tools, reviewed drafts of the paper.
- David Burstein analyzed the data, contributed reagents/materials/analysis tools, prepared figures and/or tables, reviewed drafts of the paper.
- Cindy J. Castelle reviewed drafts of the paper.
- Alexander J. Probst reviewed drafts of the paper.
- Brian C. Thomas contributed reagents/materials/analysis tools, reviewed drafts of the paper.
- Kenneth H. Williams reviewed drafts of the paper.
- Jillian F. Banfield conceived and designed the experiments, performed the experiments, analyzed the data, contributed reagents/materials/analysis tools, wrote the paper, prepared figures and/or tables, reviewed drafts of the paper.

## DNA Deposition
The following information was supplied regarding the deposition of DNA sequences:

The five Peregrinibacteria genome sequences have been deposited at DDBJ/EMBL/GenBank under the accession numbers CP013062–CP013066.

## Data Deposition
NCBI Sequence Read Accession (SRA).

SRX1085362

SRX1085365

SRX1085368

SRX1085370

SRX1085371

## Supplemental Information
Supplemental information for this article can be found online at http://dx.doi.org/10.7717/peerj.1607#supplemental-information.

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
