# Peer review of "Analysis of five complete genome sequences for members of the class Peribacteria in the recently recognized Peregrinibacteria bacterial phylum"

_PeerJ, doi:10.7717/peerj.1607_

## Round 0.1 · original submission · Major Revisions

Although both reviewers were enthusiastic about this manuscript, both also raised shared important concerns about clarity about the methods and about nomenclature of the strains. These concerns should be directly addressed in a resubmission, with enough information given for other researchers to accurately recapitulate the methods. It would also be beneficial to the manuscript if the authors included a brief section that justified the nomenclature of these strains in the context of points raised by reviewer 2.

Additionally, regardless of resubmission, I cannot accept this manuscript until the genome sequences have been made publicly available.

·

Basic reporting

The article is clear and well-written. It is a straightforward and well-reasoned analysis of some interesting and novel genomes.

A few specific points below:

The paper should not be considered for publication until the Accession numbers of the assemblies are live at GenBank. I’m unsure if PeerJ has a policy in this regard, but I personally feel that papers shouldn’t even be submitted without this information. Ideally there would also be information regarding access to the raw data.

Line 77: Do the authors really have centimeter level accuracy on the altitude of the study site?

Lines 107-108: Would be good to mention the length of the 16S rRNA gene sequences used in these analyses.

There are some formatting issues in the References that should be fixed. In particular references 1, 3, 4, and 5 need some work.

Experimental design

The results presented are really quite interesting, I’m excited that the authors are able to get closed bacterial genomes from Illumina data, much less from a metagenomic assembly. However, the description of the methods in this regard is sorely lacking. The Materials and Methods sections need to be significantly expanded.

For example, reads were trimmed with Sickle, then assembled with IDBA-UD and this was followed by manual curation. Questions that come to mind:

Is this all one sample or are there multiple samples? Are the different samples co-assembled or assembled separately? I assume binning took place… before or after assembly? How was binning accomplished? What read length was used for sequencing?

The one sentence on manual curation is particularly vague. How were assembly errors corrected? Were the reads mapped back to the assemblies? What information was used to “fill internal gaps”? I have looked at hundreds of Illumina-sequenced bacterial genomes and none have ever been close to finished because the read length of Illumina data is not long enough to span repeats in the genome… how have the authors bridged these sorts of gaps between contigs/scaffolds?

Figure 1 (Phylogenetic Tree) is also lacking in sufficient detail. How was the alignment made and what sequences were chosen for the alignment? What program was used to build the ML tree? How were the trees rooted? What genes were used in the concatenated tree?

Validity of the findings

There are two major findings in this paper, the first is the assembly/validation/annotation of these novel genomes and the second is an analysis of the metabolism/biochemistry of the organisms. I’m not much qualified to comment on the latter, but am comfortable with the results presented for the former. Though, as discussed above I would like to see much more detail presented about the actual methods. Specific comments below:


1: It’s not clear to me how the authors decided that these organisms fall into a novel Class, as opposed to an Order or Family. Since there are no cultured organisms in this entire phylum, none of the standard taxonomic rules apply. I would like to see a bit of discussion on how this determination was made.

2: The assertion that these are complete genomes seems to rest on the circularization and the GC skew plot? What about an analysis of single-copy marker genes?

3: On a similar note, the authors mention finding only single copies of the 5S rRNA, 16s rRNA and 23S rRNA genes. Did the authors look at coverage disparities in these genes to ensure that multiple rRNA gene copies were not simply collapsed in the assembly?

·

Basic reporting

There is some need for more context. Please see Comments for the author.

Experimental design

There is need for more description of the experimental design and some important methods need to be included. Please see Comments for the author.

Validity of the findings

Please see Comments for the author.

Additional comments

This is a good first look at a new set of PER genomes assembled from metagenomic data that appear to be phylogenetically distinct from the previous versions recovered from this phylum. This separation in phylogeny seems to be mirrored by some variations in genome content and metabolic potential (e.g., no Rubiscos), however, the comparisons between these new genomes and the previously published versions are limited. In general, this is a relatively superficial analysis compared to other work from this group, which is striking because they are the discoverers of, and world experts on, the PER phylum. I feel such a perfunctory description is missing a great opportunity for a comprehensive comparative genomics paper on the phylum. There could be much more context for each of the various pathways and capabilities described, especially with regards to how are these distributed among the other genomes, and how this may or may not relate to their phylogeny/evolution. That said, I know there are many reasons for needing to push manuscripts out, and I also don't believe that peer review should be used to force an altogether different version of a manuscript on authors. Thus, I will not prevent publication if the authors do not wish to do a more thorough investigation. However, I still have several concerns that I would prefer to see addressed before publishing, mostly with regards to methods, data reporting, and clarity.

Many times I’m not clear whether the manuscript is referring to a single genome or all of the genomes in the group. In some cases this is because the writing itself indicates only one genome is being discussed (line 219). Please make it clear in the text when you are referring to one genome vs. qualities shared by all.

Please provide more context for the purpose/motivation of this study. Was this part of another project? I doubt the entire study was carried out to get five PER genomes assembled to near completion. How were these five genomes selected from available data? Were they the only PER genomes in the dataset? Were they the only complete ones? I.e., how were these chosen for analysis?

Was any binning used to separate these genomes away from others? If so, this needs to be included in the methods. Also, please provide some references or more detailed protocols for how “the assemblies were manually curated to remove assembly errors, fill internal gaps, and circularize the genomes.”

There are no methods for the phylogenetic trees. How were these constructed? What alignment methods did you use? Was there any culling? Tree inference algorithms/programs (please provide settings used as well)? How were the genes concatenated? What rate and substitution models were used? What outgroup sequences were included?

Naming conventions typically give lower taxonomic designations with a similar prefix to the phylum only for the first discovered version of the taxon, e.g. the Acidobacteria phylum is named after the first isolate, Acidobacterium capsulatum. In this case, the first discovered versions of the PERs were the genomes from Wrighton et al. 2012. If the data supports the current genomes being in a new class, they should be given a different class/family/genus name, separate from Peri-, and the genomes from Wrighton et al. should be given the Peri- prefixes. I'm not sure if the authors are planning on Peribacter (current study) being separate from Perigrinibacter, or something of the like, that could be applied to the Wrighton et al. genomes, but even if this were the plan, the similarity would create a lot of confusion so I recommend these new genomes have a different prefix altogether.

Along those lines, average amino acid identity would be useful to delineate taxonomic levels too, and there is good literature on this, most notably by Konstantinidis and Tiedje.

The GenBank (or other public database) numbers for the assemblies and the raw sequencing data should be provided.

Is Table S1 only depicting one genome? What is it referring to? Please provide a legend.

Why is Figure S1 not referenced in the manuscript?

Please clarify “primarily” on line 120. What else does the variation involve?

Why is the data intriguing on lines 195-197? Please explain.

Which genome are is being referred to on line 219?

Please clarify what is meant by “nearly syntenous” on line 223.

In the Figure 1 legend, it says bootstrap values are marked on the nodes, but not for all nodes. What is special about the ones that aren’t marked (low support?). Also, if there were 1000 bootstrap runs, why is the maximum value 100? This should read percent of bootstrap runs, I believe, or otherwise, report true numbers.

Please provide more information for Figure 2, such as what the colors represent, what each of the alignments are showing, etc. Currently the legend does not facilitate interpretation of the figure, and it is unclear how the statement at the top of Figure 2 relates to the sentences in the text where it is referenced.

---

## Round 0.2 · accepted · Accept

Thank you for addressing the reviewers critiques adequately, and apologies on the slight delay over the holidays. I'm happy to accept this manuscript for publication.